# Spatial Variability and Detection Levels for Chlorophyll-a Estimates in High Latitude Lakes Using Landsat Imagery

**Filipe Lisboa** [1,*] , **Vanda Brotas** [1,2] , **Filipe Duarte Santos** [3] , **Sakari Kuikka** [4] , **Laura Kaikkonen** [4] **and Eduardo Eiji Maeda** [5]

1   MARE—Marine and Environmental Sciences Centre, Universidade de Lisboa, Campo Grande, 1749-016 Lisboa, Portugal; vbrotas@fc.ul.pt
2   PML—Plymouth Marine Laboratory, Prospect Place, Plymouth PL1 3DH, UK
3   CCIAM-CE3C, Faculty of Sciences, University of Lisbon, Tapada da Ajuda, 1349-017 Lisboa, Portugal; fdsantos@fc.ul.pt
4   Faculty of Biological and Environmental Sciences, Ecosystems and Environment Research Programme, University of Helsinki, 00014 Helsinki, Finland; sakari.kuikka@helsinki.fi (S.K.); laura.m.kaikkonen@helsinki.fi (L.K.)
5   Department of Geosciences and Geography, University of Helsinki, P.O. Box 64, 00014 Helsinki, Finland; eduardo.maeda@helsinki.fi
*   Correspondence: fblisboa@fc.ul.pt

**Abstract:** Monitoring lakes in high-latitude areas can provide a better understanding of freshwater systems sensitivity and accrete knowledge on climate change impacts. Phytoplankton are sensitive to various conditions: warmer temperatures, earlier ice-melt and changing nutrient sources. While satellite imagery can monitor phytoplankton biomass using chlorophyll a (Chl) as a proxy over large areas, detection of Chl in small lakes is hindered by the low spatial resolution of conventional ocean color satellites. The short time-series of the newest generation of space-borne sensors (e.g., Sentinel-2) is a bottleneck for assessing long-term trends. Although previous studies have evaluated the use of high-resolution sensors for assessing lakes' Chl, it is still unclear how the spatial and temporal variability of Chl concentration affect the performance of satellite estimates. We discuss the suitability of Landsat (LT) 30 m resolution imagery to assess lakes' Chl concentrations under varying trophic conditions, across extensive high-latitude areas in Finland. We use in situ data obtained from field campaigns in 19 lakes and generate remote sensing estimates of Chl, taking advantage of the long-time span of the LT-5 and LT-7 archives, from 1984 to 2017. Our results show that linear models based on LT data can explain approximately 50% of the Chl interannual variability. However, we demonstrate that the accuracy of the estimates is dependent on the lake's trophic state, with models performing in average twice as better in lakes with higher Chl concentration (>20 μg/L) in comparison with less eutrophic lakes. Finally, we demonstrate that linear models based on LT data can achieve high accuracy ($R^2$ = 0.9; *p*-value < 0.05) in determining lakes' mean Chl concentration, allowing the mapping of the trophic state of lakes across large regions. Given the long time-series and high spatial resolution, LT-based estimates of Chl provide a tool for assessing the impacts of environmental change.

**Keywords:** Chlorophyll-a; phenology; landsat; high-latitude lakes; climate change

## 1. Introduction

High latitude lakes are considered sentinels for a climate in change, due to their coupling with ice phenology, response to changes in humidity and precipitation patterns, and the high sensitivity of photosynthetic organisms to changing temperatures [1,2]. In particular, the growth rates, abundance, and species composition of phytoplankton are an indicator of changing environmental conditions [2–4]. Nevertheless, monitoring water bodies has been difficult because in situ methods are very localized and cannot always be performed routinely. In situ datasets might also contain data gaps, and data gathering is not coherent across different regions and field sampling is deemed expensive.

A powerful way to have synoptic views of changes in water bodies is to monitor them from space. Phytoplankton biomass, which can be proxied from the chlorophyll *a* (Chl) concentration, can be used to estimate primary production in aquatic environments. Satellites have now been collecting decades of remotely sensed optical imagery over large areas. Due to the characteristic reflectance of Chl pigments, we can estimate concentrations from such images. Earth Observation satellites have been designed and launched with the specific purpose of studying phytoplankton from space with sensors specifically suited to the assessment of aquatic ecosystems. For example, NASA's Sea-viewing Wide Field-of-View Sensor (SeaWiFS), launched in August 1997 onboard the SeaStar satellite, collected data until 2010 at a resolution of 1.1 km. The Terra and Aqua Satellites both collect data through the 36-band MODIS sensor at wavelengths between 0.41 and 14.24 µm applicable to extensive ocean color algorithms [5]. Additionally, the MERIS sensor was an imaging spectrometer at fifteen spectral bands and a spatial resolution of 300 m over land. Its visible to near-infrared sensitivity ranged from 390 nm to 1400 nm at programmable bandwidths between 2.5 and 30 nm. Following the loss of the Envisat-1 payload in 2012, the MERIS data spans until April 2012, having started in May 2002. Recently, the European Space Agency (ESA) has launched the Sentinel 3 constellation as part of the Copernicus Programme. Onboard the two sentinels, the Ocean and Land Colour Instruments (OCLI) are collecting data at wavelengths from 0.4 µm to 1.02 µm at 21 spectral bands, allowing for algal pigment discrimination and the further development of phytoplankton functional types characterization from space [6]. The resolution of the new OCLI sensor is 300 m, which is not suitable for studying lakes smaller than 30 ha.

The use of the above-mentioned satellites hampers the study of small lakes due to their coarse resolution in relation to the size of the lakes. In fact, 95% of the roughly 56,000 Finnish lakes have an area <1 km$^2$ [7]. Additionally, consistent data from the Ocean Colour missions' dates to 1997, which provides a relatively short time series. Given these limitations, the high-resolution Landsat satellites (LT) can be advantageous to use on small water bodies. LT satellites have collected one of the most comprehensive datasets, with almost 40 years of observations. LT 5 was launched in March 1984 providing data for the following 28 years. LT 5 carried the 7-band Thematic Mapper (TM) multispectral sensor. LT 7, launched in April 1999, is still in operation and carries the 8-band Enhanced Thematic Mapper Plus (ETM+). Exploring this archive for assessing Chl concentrations offers an excellent opportunity to evaluate how changes in phenology (i.e., timing of the blooms and their duration) have occurred in previous decades, due to both climate change and eutrophication. Furthermore, the high spatial resolution of LT data (30 m) allows the assessment of smaller lakes or lakes with complex shapes, minimizing the influence of surrounding land vegetation.

Given Finland's geographical location, the country can experience a faster warming with respect to the global average. Warming in Finland is estimated to be roughly 50% higher than the global average, which corresponds to a 0.15–0.20 °C increase per decade [2]. Most Finnish lakes are from glacial origin and during most winters the lakes in Finland have ice cover, resulting in an annual cycle of water quality parameters that also affect primary production dynamics [8].

Phytoplankton phenologies vary within the lake's trophic state and climate induced changes in the environmental conditions affect both the species composition and dynamics of the phytoplankton community [9]. In many lakes, the effects of eutrophication are common with algal blooms being exacerbated by climate change [3], it is therefore critical to have extensive temporal and spatial information on Boreal lakes [10].

The LT data has been used for the assessment of regional lake water clarity with reliable empirical relationships between satellite data and ground observations in the north American glaciated lakes [11] and in Europe [12]. Similarly, the interest on studying Finnish lakes through LT data dates back to the 1980s, leading to research that proved that spectral differences due to aquatic vegetation were possible to detect [13]. An important early study on Finnish lakes also approached the configurations of ETM+ sensor for lake water quality estimations [7]. Feasibility studies conducted in Finland in the early 2000s identified that it is possible to estimate the average Chl level of an individual lake through matchups of airborne sensing data from another lake [14]. Notwithstanding the limitation on spectral and radiometric resolutions, other previous efforts outlined the importance of ETM+ data on monitoring small lakes for colored dissolved organic materials (CDOM), turbidity and Secchi disk transparency [15]. We find that these approaches should be revisited with LT data for two reasons, first, taking the advantages of long data series is paramount for studying change [16], secondly, exploring alternative high-resolution imagery is a necessary step to enhance the monitoring of small water bodies. Previous studies used semi-empirical approaches to conclude that Chl assessment is limited by the band configuration of LT, and this type of activity goes beyond the design of the TM sensor [17]. Our study revisits this assumption by gathering more in situ data and a vast collection of imagery.

Studies of single lakes using LT data are well described in the literature [12,18–20]. For example, imagery from LT and samples from a transect were combined achieving a $R^2 = 0.9$ over Lake Erken, Sweden [12]. Another study [21] used the band ratio of B2/B1 (green/blue) for Chl assessment on Lake Champlain, USA, reporting an R of 0.82. Having such space-derived estimates has been proved useful to study the changing patterns [22]. Nevertheless, the challenges increase with smaller lakes. For example, a set of 131 small lakes (averaging 100 ha) in Maine revealed an $R^2$ of 0.25 [23]. LT5-TM and LT7-ETM+ are not ideal for waterbodies, but they are the only available dataset at enough resolution with a significant temporal span.

The models derived in these previous studies were limited by validation campaigns carried at specific validation campaigns, in specific time frames. Our approach is to engage on an innovative in situ and remote sensing match up. We used many years of consistent in situ data, collected at different locations and created a validation method, using more than three decades of satellite data. To take full advantage of the long LT time series, models should robustly incorporate data from both LT5-TM and LT7-ETM+. Vicent et al. argued that a model developed for LT5 cannot be applied to LT7 images [24]. Nonetheless, their methods involved the use of dark-object-subtracted radiance instead of surface reflectance. Hence, further studies are still needed to fully explore the combined use of LT5-TM and LT7-ETM+, using surface reflectance derived from state-of-the-art atmospheric correction algorithms [25].

The focus of our study is to demonstrate whether Chl assessments using surface reflectance retrieved from LT sensors can be generalized for these different circumstances. In other words, we aim to understand how different Chl might affect LT reflectance, as a preliminary approach to develop advanced bio-optical models for inland and coastal waters. Such models, in turn, are an essential tool not only for global water quality monitoring, but to also study temporal dynamics of phytoplankton across large areas, allowing the identification of shifts in the temporal characteristics of phytoplankton blooms related to climate variability and land-use changes [26].

The water-leaving signal is especially complex for the case of inland waters due to the high concentrations of CDOM, tripton, floating and submerged aquatic vegetation, suspended matter (e.g., clay) and finally phytoplankton. The tripton component, composed of detritus and minerals from dead phytoplankton and decaying organic matter, varies significantly in optical signatures and concentration ranges. High concentrations of humic and fulvic acids from surrounding vegetation are leached into lakes and reduce the light availability in the blue spectral region in boreal and brackish waters [26], both being the case of most of Finland's lakes. The lakes considered in this paper are, thus, some of the most difficult cases for bio-optical models of phytoplankton proxying from space.

Our objective is to better understand spatiotemporal factors affecting the relationship between LT reflectance data and surface Chl concentration over high latitude lakes. We drive our research by asking: 1) how does the performance of models based on LT data vary with different lakes' trophic state? 2) How does the performance of models based on LT data vary at different temporal scales? 3) Can the LT data archive be used to assess the spatial and temporal variability of lakes' average Chl levels across large areas?

## 2. Material and Methods

### 2.1. Study Area and In Situ Data

The lakes included in this study belong to the intensively monitored lakes of the national routine monitoring network of Finland (Figure 1). Lake Saimaa, along the Salpausselkä system, is the only lake with two sampling spots. Water samples for turbidity were taken from 1 m depth, and Chl was determined from a composite sample of 0–2 m depth. The concentration of Chl was measured with a spectrophotometer after extraction with hot ethanol (ISO 10260 (Water quality-Measurement of biochemical parameters–Spectrometric determination of the chlorophyll a concentration. International Organization for Standardization, Geneva, 1992.), GF/C filter). Turbidity, measured in Formazin Nephelometric Units, was determined by the nephelometric method (EN 27027 (Water quality. Determination of turbidity. European Committee for Standardization, Strasbourg, 1994.)), based on measurement of light (860 nm) scattered within a 90° angle from a beam directed at the water sample, with formazine used as a standard matching solution.

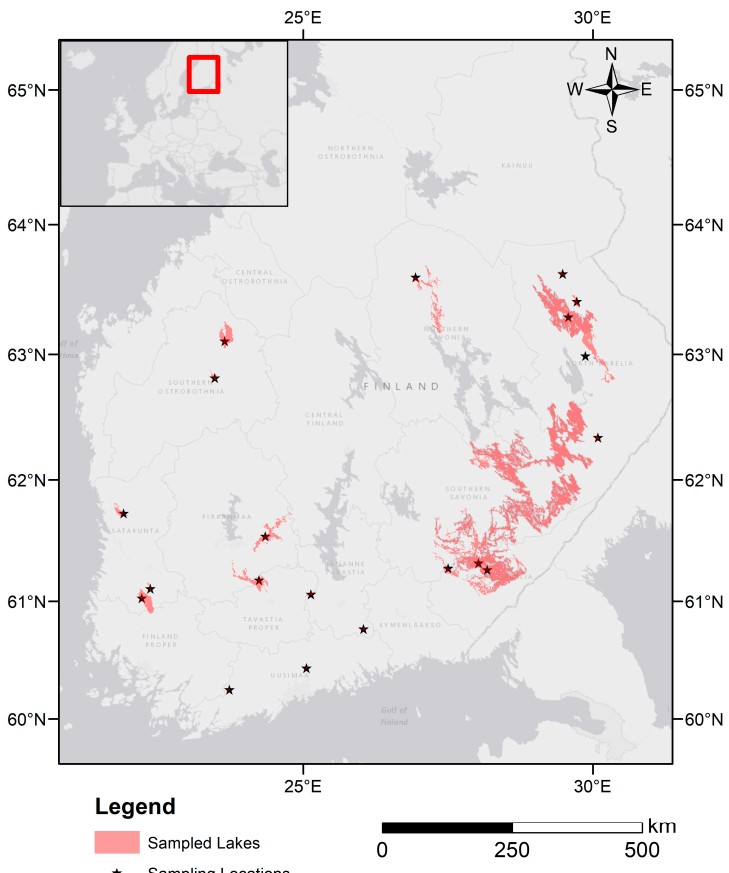

**Figure 1.** Distribution of the 19 sampled lakes and the 20 sampling locations. Two sampling locations were defined for lake Saimaa, the biggest lake in Finland.

### 2.2. Landsat Imagery

We used the Surface Reflectance Collection—Tier 1 from LT 5 and 7 provided by the United States Geological Survey (USGS) to the Google Earth Engine archives. Both satellites' lifespans include all our in situ sampling dates. These products are processed through the Landsat Ecosystem Disturbance Adaptive Processing System (LEDAPS) atmospheric correction—a radiative transfer model that includes water vapor, ozone, geopotential height, aerosol optical thickness and digital elevation [25]. Only images with the highest quality factor were used summing a total of 3865. Images are predominantly from summer months; no images were selected from December and January, due to high cloud coverage and low-or null-sun elevation. The dates of the images span from February 1999 to April 2017 (Figure 2). We have used bands 1–5 from the LT collection corresponding to all visible bands: near infrared (NIR-band 4) and shortwave infrared (SWIR -band 5).

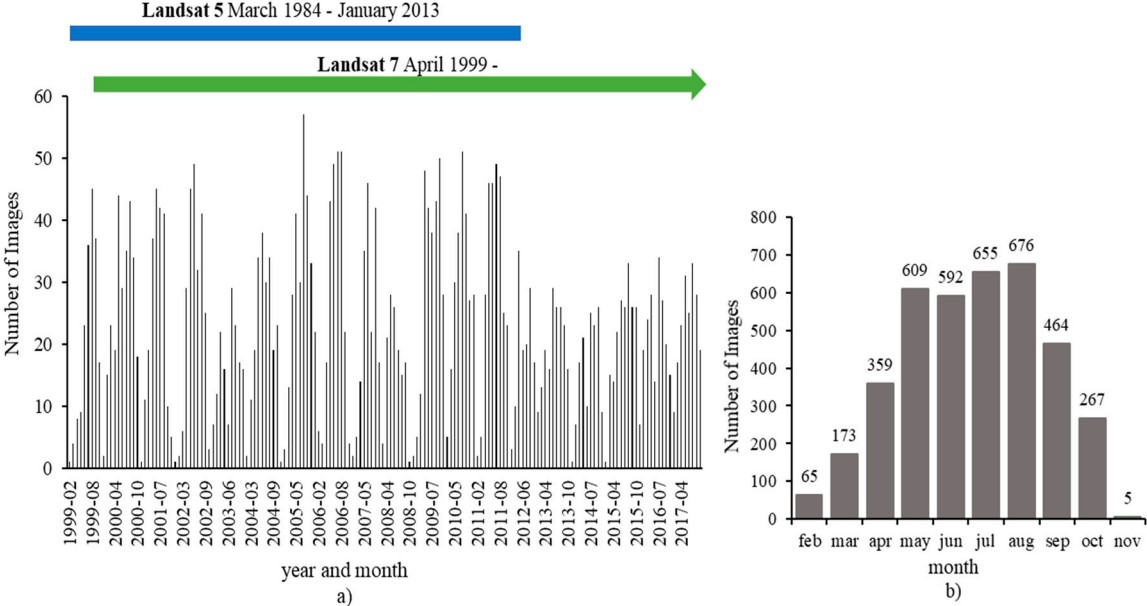

**Figure 2.** Number of images used for this study throughout the whole imagery collection (**a**) and the distribution per month (**b**).

### 2.3. Data Methods

We evaluated multivariate regression models to identify which LT bands, or band combinations, are more suitable for estimating Chl concentrations. We also evaluated how time differences between the date of Chl field samples and the date of LT images acquisition affect the performance of the models.

Our code was developed in Google Earth Engine [27], which provides the entire dataset from LT imagery and the possibility to define the regions of interest according to our specific needs. The collection of LT data into band reflectance time series over the lakes was carried through two main stages: quality assurance of the images and matching of spatially averaged water reflectances with the in situ samples.

Next, we gathered and filtered the image collection. The filter removed cloud cover, cloud shadowing and ice pixels, all masked out based on the C Function of Mask (*cfmask*) algorithm described by [28]. Nevertheless, the *cfmask* cannot account for all imagery artefacts. To solve the issue, we added a normalized difference chlorophyll index (NDCI) to the resulting LT collection. We calculated the 25th and 75th percentiles from the resulting index distribution, thus eliminating pixels with NDCI outside the interquartile range (IQR) and bullet-proofing the *cfmask* filtering.

**Daily Aggregated Data**

Daily aggregated data is relative to the match up of a satellite image with the in situ collection around that date. We spatially averaged the resulting valid pixels per image, as laid down in the area buffer defined by a permanent water mask. We gathered data according to the temporal averaging of images collections within ± $n$ days from the date of in situ collection. Previous validation studies indicated an ideal time gap between satellite overpass and field measurement of ± 1–2 days [29]. We have chosen a time gap of maximum ± 1–2 days, from the hour of sampling, to estimate the Chl concentration of all the 19 inland water bodies.

**Seasonally Aggregated Data**

Our analysis also evaluated seasonally aggregated data, in which we lose the temporal aspect of remote sensing concentration to assess the spatial distribution of the lakes' trophic status. For this, we used the average Chl during spring and summer (day of year [DOY] from 100 to 280) in all lakes, considering all LT pixels within a 500 m buffer of sampling collection. This procedure resulted in a total of 126 validation pairs.

All procedures were carried out in GEE and the data exported for further statistical analysis in R [30] as detailed in Section 2.4. For statistical purposes, in situ Chl data was primarily filtered to remove outliers. To detect these outliers, we used Tukey's method which targets values above and below 1.5 of the interquartile range (IQR). This study also includes an assessment of the relationship between turbidity and Chl. We calculated the overall correlation between both variables and narrowed it down to summer data (DOY 200 to 280).

**Defining a Permanent Water Mask.**

It is paramount that the spatial average of water-leaving reflectance values is performed over permanent water surfaces. Preliminary lake boundaries were defined according to the GLOWABO dataset [31]. However, this layer can include small lake islets or areas of intermittent water presence. To further ensure that the imagery corresponds to permanent water surfaces, we intersected the GLOWABO dataset with the Global Surface Water Data from a Joint Research Center study [32]. By intersecting these datasets, we ensure that the selected pixels correspond to permanent water surfaces in the period between 1984 and 2017. Our methodology results in the definition of detailed and high-resolution surfaces on which reflectance spatial averaging can be performed. Within these areas, we overlaid a circle, centered around the sampling location, with varying a radius (Figure 3). Additionally, to avoid interference from land vegetation and other types of aquatic plants, we applied a buffer of 50 m away from land when the circle of extraction is above 500 m.

All shaded areas on the map of Figure 1 depict the surfaces over which the reflectance bands were collected in this manner. After creating a time-series collection of surface reflectance for each lake, we generated a database for in situ Chl match-up and derived the models.

*2.4. Model Selection through Relative Importance Metrics*

We assessed the performance of a linear model by merging data from both LT 5 and 7 and all lakes. Multiple linear fits are presented in this analysis, providing a measure on how the different band combinations perform on detecting Chl. To have an overview of the variables to incorporate into the model, we used the "Leaps: regression subset selection" exhaustive search, the routine for the best subsets of predictors which uses a linear regression through an efficient branch-and-bound algorithm. The resulting combinations of regressors can be analyzed by either the $R^2$ of the model or its Schwartz's information criterion or Bayesian information criterion (BIC). The BIC is a criterion for model selection among a finite set of models [33]. While fitting models, it is possible to increase the likelihood of a good fitting by adding parameters but doing so may result in overfitting. The BIC resolves this problem by introducing a penalty term for the number of parameters in the model. Lower values of BIC are then preferred for the best model assessment.

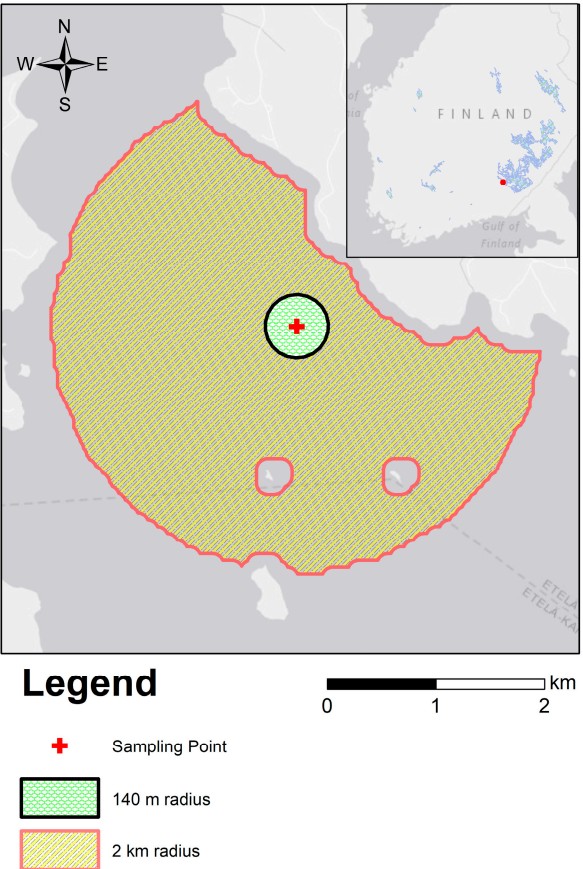

**Figure 3.** One of the sampled locations at the center of the area for satellite reflectance band extraction (shaded). Such areas are a variable radius circle around the sampling site and avoiding land.

Multivariate models for Chl were designed using each reflectance band as a regressor. Secondly, we evaluated model performance through relative importance metrics. Relative importance metrics is the quantification of the individual regressor's contributions to a multivariable regression model. For uncorrelated predictors, the multivariate coefficient of determination ($R^2$) is simply the cumulative result of each $R^2$ of the single-variate linear model. However, bands of satellite optical imagery are deeply correlated, raising the necessity for a relative importance classification. We have used the *relaimpo* package from R that provides six different methods for assessing relative importance in linear regression [34]. Among these methods, the one from Lindeman, Merenda and Gold (LMG) [35] is one of the most computationally intensive but highly recommended.

### 2.5. Spectral Indices

Previous studies have demonstrated the usefulness of spectral indices, rather than individual bands, on the detection of vegetation both in land and water. In addition to the individual bands, we used two spectral indices to derive our models; the LT adapted version of the normalized difference chlorophyll index (NDCI), and another index, the blue-red-green (BRG), based on the relative difference of blue and red bands relative to the green band.

The NDCI is given by:

$$\text{NDCI} = \frac{\rho_{NIR\ (B4)} - \rho_{red\ (B3)}}{\rho_{NIR\ (B4)} + \rho_{red\ (B3)}} \tag{1}$$

where $\rho$ is the reflectance of the specific band.

The BRG model was adapted from [19] who achieved an $R^2 = 0.818$ in Lake Garda, Italy. In such a model, Chl is directly proportional to the reflectance index given by:

$$BRG = \frac{\rho_{blue\ (B1)} - \rho_{red\ (B3)}}{\rho_{green\ (B2)}}$$

(2)

The same BRG index led to satisfactory results for other small water bodies as in the Malilangwe Reservoir, Zimbabwe with $R^2 = 0.81$ [36].

## 3. Results

### 3.1. Daily Aggregated Data

**Model and Variable Selection**

Of all models derived through intra-annual data, the best performance was achieved with a buffer of 700 m and a time-window of ±1 day from the satellite acquisition. For this configuration, the results of the LEAPS exhaustive variable search are presented in Figure 4. The $R^2$ varied from 0.16, using only one explanatory variable (band 4 - NIR), to 0.55, using all possible bands and indices. The highest $R^2$ value could also be achieved when excluding band 2 (green) and including all other variables. The BIC values varied from -30, using only band 4, to −150, using bands 1, 3, 4 and 5.

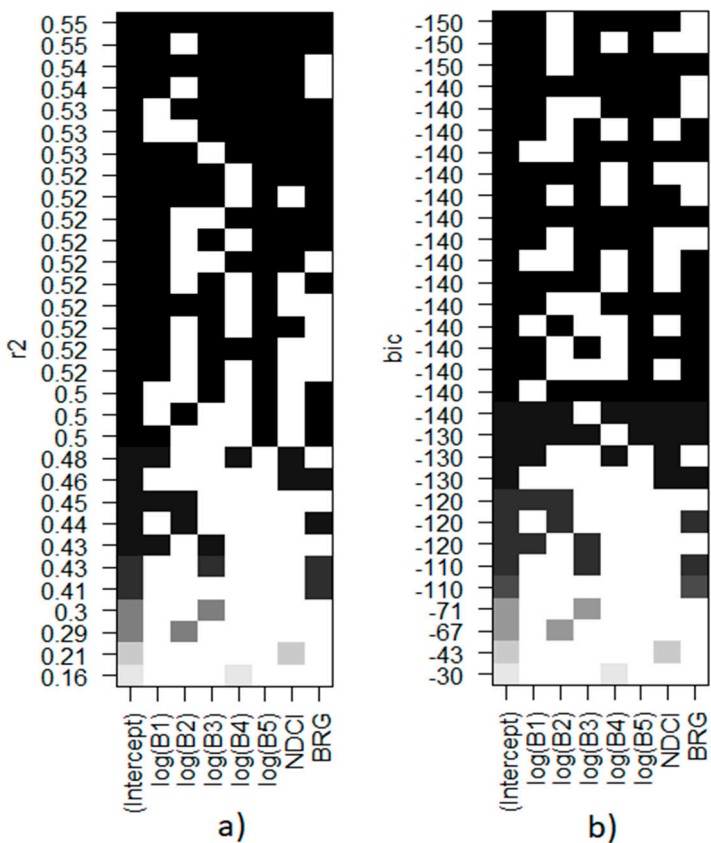

**Figure 4.** Best linear model assessment through LEAPS variable search. In (**a**) the $R^2$ for the models for the different regressors combinations. In (**b**) the Bayesian information criterion (BIC) for the respective models.

Figure 5 depicts the relative importance of each band for the overall $R^2$ of the achieved model. All bands 1–4 are significant ($p$-value < 0.05) for the construction of the model, although the different models used to evaluate their relative importance did not always agree on the most important variables.

Across all methods, band 3 and BRG index has the strongest explanatory power. Band 2 was selected as the second-best explanatory variable by two methods (LMG and First).

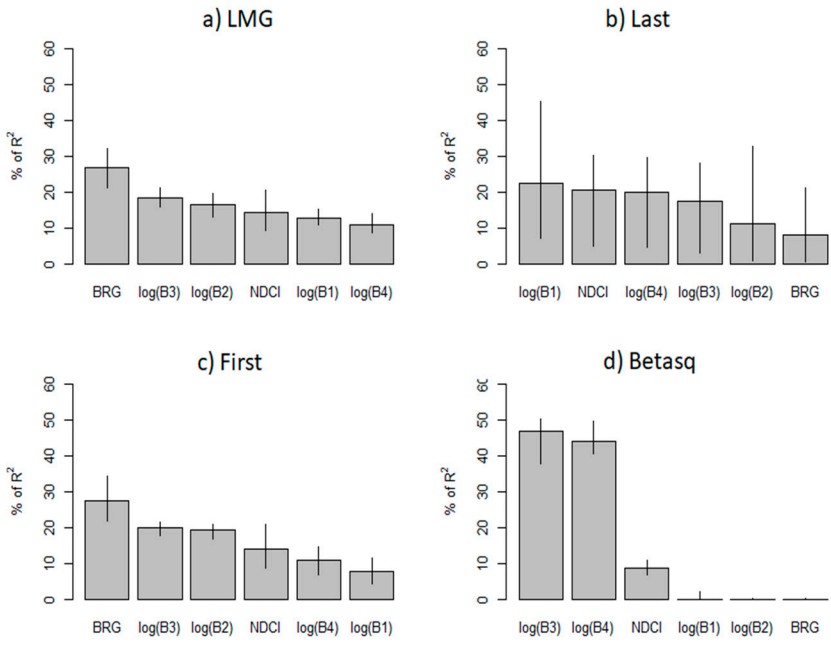

$R^2 = 52.01\%$, metrics are normalized to sum 100%.

**Figure 5.** Metric of relative importance using the 700 m radius and a time window of ± 1 day. All bands and the spectral indices included. Relative importance of each Landsat (LT) band with 95% bootstrap confidence intervals.

Figure 6 provides the correlations matrix of all bands and the NDCI and BRG indices. As evidenced, relationships between individual explanatory variables and Chl are not linear. Pearson's coefficient of correlation (R) is shown for each band, and indices, with relation to the Chl concentration.

As a result of the small contribution of the shortwave infrared (SWIR) band 5, we considered a final model for the daily aggregated data comprising all these bands and indices.

All individual bands 1–5 correlate positively with in situ Chl, but bands 2 (R = 0.53) and band 3 (R = 55) are more strongly correlated. Both indices NDCI and BRG are negatively correlated with in situ Chl.

**Model Performance**

As expected, shortwave infrared (SWIR) band 5 of LT 5 and 7 provided only a marginal contribution to the overall performance of the model by 0.04 in the $R^2$ coefficient. Extracting that band to assess the performance of only visible and NIR bands also produced important results, summarized in Table 1. All models derived from single bands 1–4 were significant, but not all models that included reflectance indices, NDCI and BRG, are significant. Henceforth, for the best models, we checked if all coefficients are significant, i.e., we checked if all variables significantly contribute for the resulting model. For the best performing model, a $R^2 = 52.01$ was achieved, including bands 1–4 and both indices. Reducing the time-window between satellite collection and in situ sampling has a relevant impact on the overall performance of the model, as it increases the $R^2$, whilst maintaining a *p*-value < 0.05.

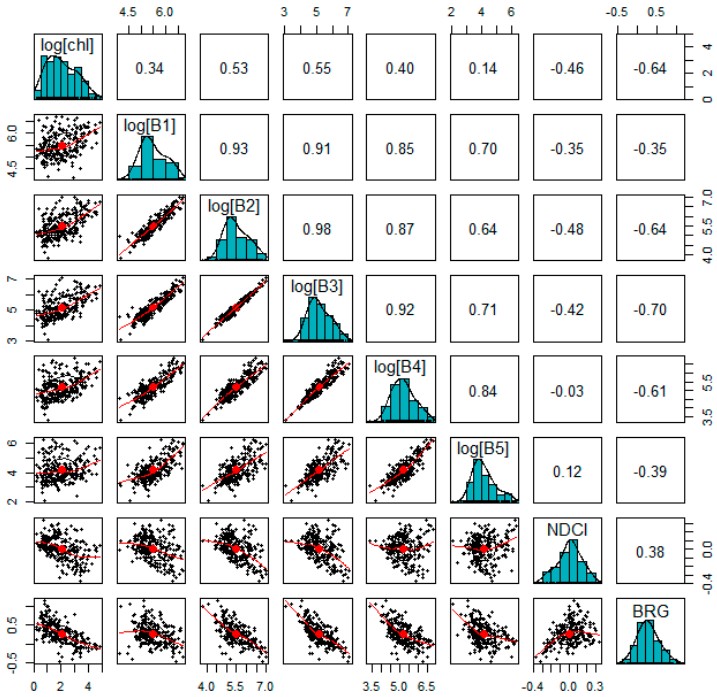

**Figure 6.** Scatterplots matrix with field measured chlorophyll a (Chl) and LT bands the numbers on the diagonal part are the coefficients of correlation, R. The red lines depict polynomial fits to the individual band scatterplots.

**Table 1.** Buffer size and $R^2$ for time-windows of ±1 and ±2 days. All tests were performed for the model with bands 1–4, we excluded band 5, as, in some cases, it leads to non-significant models (*p*-value 0.05). All bands were significant and all models with *p*-value < 0.05. For the models derived with the indices, non-significant regressors are shown in brackets.

| Buffer Size/t-Window | 60 m | 90 m | 100 m | 140 m | 180 m | 300 m | 500 m | 600 m | 700 m | 800 m | 900 m |
|---|---|---|---|---|---|---|---|---|---|---|---|
| | | | Model including bands 1–4: | | | | | | | | |
| ±1 day | 47.81% | 46.82% | 47.15% | 47.68% | 43.70% | 42.87% | 48.27% | 47.80% | **48.75%** | 48.7% | 48.47% |
| ±2 days | 45.15% | 37.71% | 36.95% | 37.51% | 37.34% | 38.63% | 41.4% | 41.05% | 41.13% | 41.77% | 41.95% |
| | | | Model including bands and indices: | | | | | | | | |
| ±1 day | 49.96% [Du] | 50.89% | 50.77% | 50.19% | 47.16% | 45.15% | 51.26% | 50.98% | **52.01%** | 51.66% | 51.92% |
| ±2 days | 45.87% [B2, B3, B4, NDCI, BRG] | 39.38% [B1, B2, BRG] | 38.83% [B1, BRG] | 39.89% [BRG] | 39.88% | 40.28% | 42.53% [B3] | 44.39% | 44.66% | 44.93% | 45.21% |

The choice of the size of the buffer around the sampling location has a small impact when compared to the choice of the time-windows (Table 1). It is advisable to maintain a buffer size of either 60 m or above 500 m. Between 90 and 300 m, we observed that the number of additional pixels has created more noise than valuable signal from Chl content, thus creating a lower performing model.

Figure 7 shows the best performing model found with a buffer of 700 m and ±1-day time window; this model includes all visible bands plus near infrared (NIR, band 4) and shortwave infrared (SWIR, band 5). The inclusion of SWIR slightly increases the overall correlation but also the *p*-value of the model.

**Table 2.** Individual band/index importance for the model in Figure 7.

| | log(B1) | log(B2) | log(B3) | log(B4) | log(B5) | NDCI | BRG |
|---|---|---|---|---|---|---|---|
| **LMG** | 0.110009 | 0.149998 | 0.18466 | 0.106984 | 0.075028 | 0.128362 | 0.244957 |

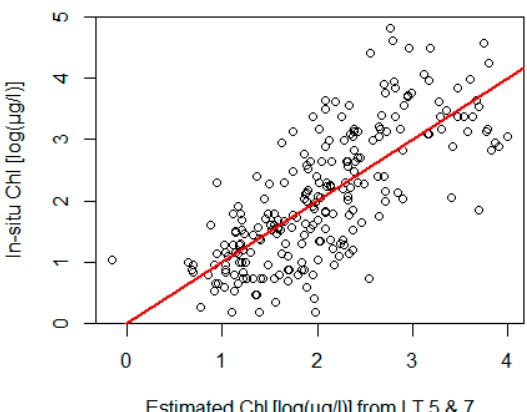

**Figure 7. Scatterplot with modelled vs. observed Chl using the best model (with R$^2$ and *p*-value).** This model was achieved in the conditions of Table 2: all five bands, NDCI and BRG for a time window of ±1 day and 700 m buffer.

Using satellite images from ±1 day of in situ sampling comes with the disadvantage of a small number of data pairs (Figure 7).

Indeed, the tests in Table 1 and the model in Figure 7 incorporate data from 6–11 out of the 19 lakes—for the remaining lakes, there are no satellite passes free of cloud cover, enough quality control standards, sun elevation or snow-free for the match-up. To assess Chl detection levels from different lakes, we have widened the time-window to ±4 days and lowered the radius to 500 m and included band 5 (SWIR). We have compared results obtained for individual lakes by addressing the lakes' characteristics like depth, size and proximity of the sampling to the lakes' margins.

Results from the individual multivariate models for each lake (Figure 8) reveal that model performance is highly dependent on the Chl level. In Figure 9, we show that the coefficient of determination, R$^2$, varies depending on the mean Chl concentration of each lake which we categorise as oligotrophic, mesotrophic and eutrophic. For each trophic state of a lake, the accuracy of the estimates from satellite imagery can vary greatly, but there are significantly better performances for eutrophic lakes.

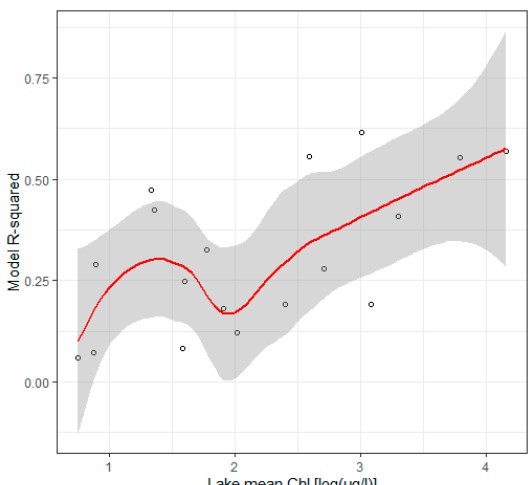

**Figure 8.** Scatterplot of individual lakes R$^2$ vs. average Chl concentration in each lake. The graph also shows the root-mean-square error (RMSE) as a shadowed area and the blue line calculated as the locally weighted least squares regression. Results were achieved using individual multivariate models for each lake and a time-window of ±4 days.

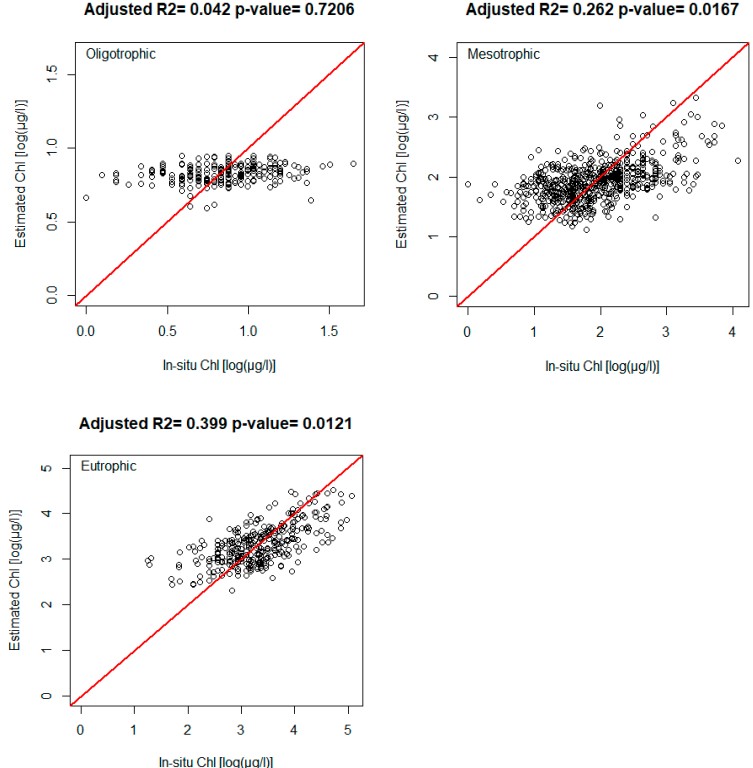

**Figure 9.** Model performance according to the trophic class of the lakes.

### Model Performance and Trophic State

The trophic state of lakes greatly influences the performance of the model built on the individual samples match-up. According to our results in Figure 9, eutrophic lakes have better-resolved Chl estimations. Our results show that for eutrophic lakes the model performs twice as better as for mesotrophic lakes. All data points in Figure 10 were filtered to the spring and summer season, i.e., DOY = [100:280].

### Model Performance Against In Situ Measurements of Chl and Turbidity

Our results show that turbidity and Chl are strongly correlated during summer months, i.e., DOY = [200:280]. Turbidity is therefore highly coupled with Chl during ~2.5 months of the year and before that the relationship is not present. The fact that during spring and until July the Chl is decoupled from turbidity provides a ground for phenology studies on phytoplankton blooms in lakes. Even in summer period, a model of LT reflectance and turbidity does not perform nearly as well as for Chl.

### Performance of the Model Vis-a-Vis Sampling Location and Geophysical Characteristics

Sample collection set-up greatly influenced how well Chl concentrations at particular locations can be estimated using remote sensing (Figure 11). Therefore, model performance is likely to vary within lakes that are close to each other. There is no linear trend relating depth and $R^2$, but the smaller the lake the better this model performs. Indeed, small lakes of less than 40 km$^2$, are better suited to be studied under this model. The relationship between the small lakes signal and the surrounding vegetation will be discussed further. Sampling locations that are very close to the shore (less than 200 m) have poor relation with the model estimates. On the other hand, the performance of the model is better for sampling locations between 200 and 700 m to the lakes' margins (Figure 11c). As will be discussed further, proximity to a lake's margins can imply higher Chl if surrounding vegetation does not interfere with the optical characteristics of a lake's water.

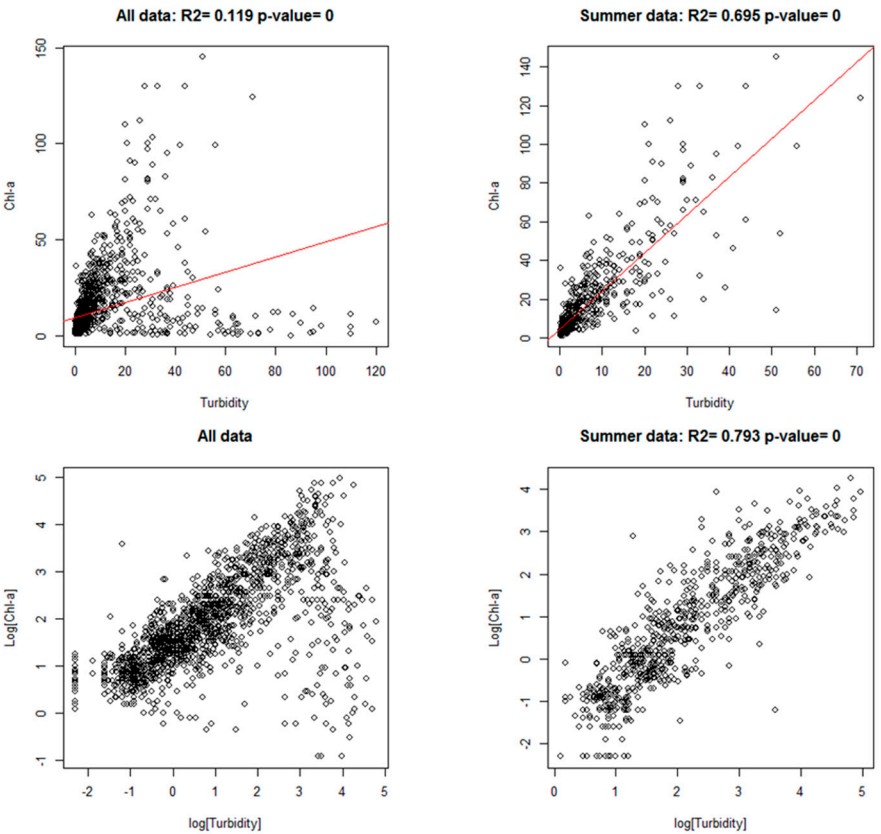

**Figure 10.** The relationship between Chl and turbidity for all data (**left**) and summer months (**right**).

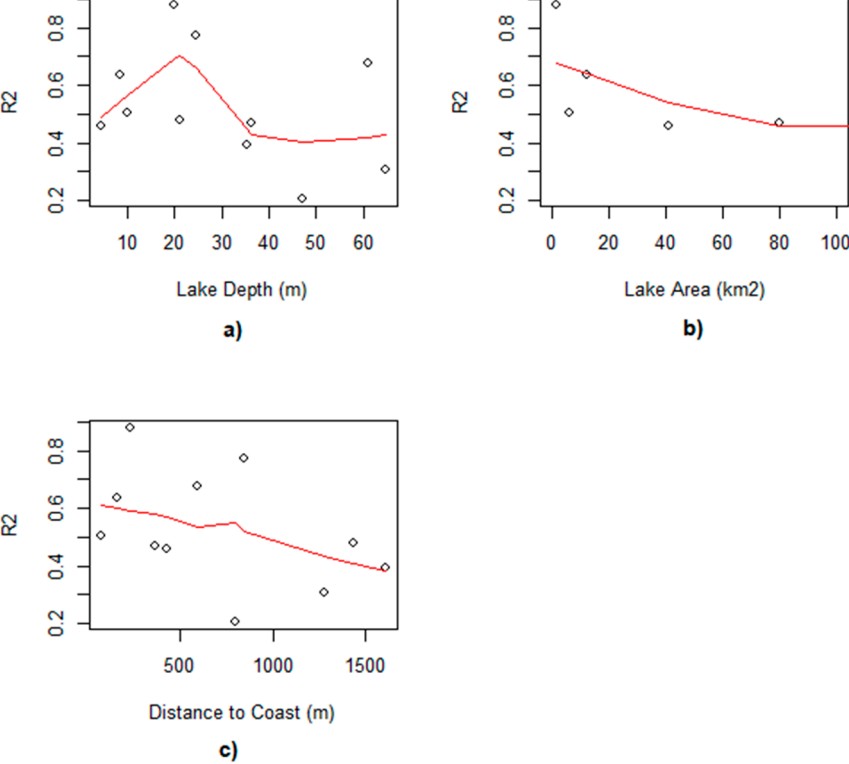

**Figure 11.** Scatterplot matrix $R^2$ vs. lake depth (**a**), lake size (**b**) and distance from the sampling collection site to the coast (**c**).

### 3.2. Seasonally Aggregated Data

We showed that the signal-to-noise ratio on the remotely sensed Chl is higher on the spring and summer period for eutrophic lakes. Hence, we aggregated images collected only during the summer period, i.e., between DOY 100 and 280. These images, when matched-up with the in situ samples, produce a well-performing model.

The seasonally aggregated model (Figure 12) is the result of using a 500 m buffer with a ±2 day with averaged summer results for each lake. This result shows that the model performs well for large-scale estimation of Chl between DOY 100 and 280. The results of the Chl estimation for the whole of Finland, using the corresponding model, is shown in Figure 13. As we expect that a gradient of rising latitudes has a large influence on the primary production of the lakes, we averaged the mean Chl in our model for the same latitude. For latitudes between 61.5°N to 63.5°N, there seems to be a clear positive trend on the increase of seasonal Chl. Lake Saimaa (red diamond in Figure 13) is one of the less eutrophic ones, which contributes to the lowest average Chl concentration at the 61.2°N parallel. On the other hand, Lappajärvi (red star in Figure 13 and detailed in Figure 14) is Finland's largest crater lake, it has been given by our model as a very eutrophic lake, which corresponds to the same concerns in the literature.

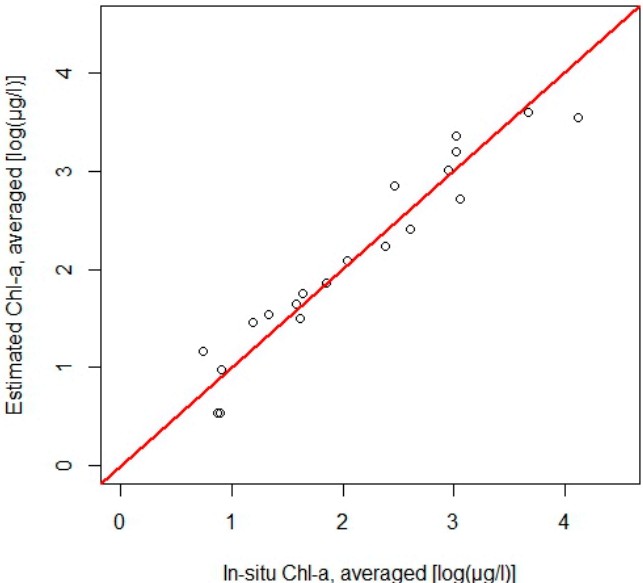

**Figure 12.** Predicted values of lakes mean surface Chl, using multivariate regression.

The clear trend contrasts with the northernmost areas where lakes are less abundant, smaller or sparser. Above 63.5°N, extreme variance of Chl can be seen. Under 63.5°N latitude, the model correctly identifies lakes that are typically eutrophic, due to the presence of agriculture or arable land.

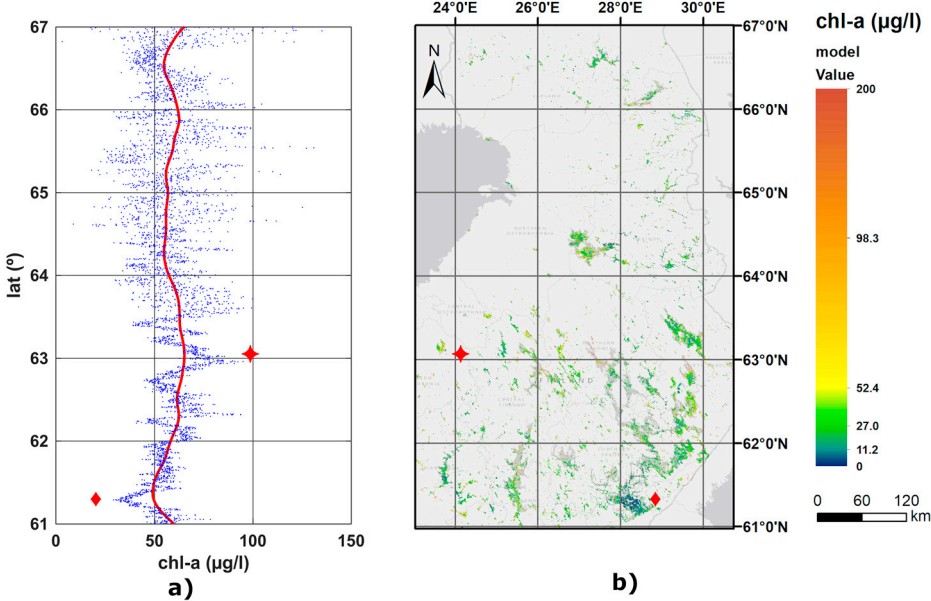

**Figure 13.** (**a**) Blue dots represent the average Chl per latitude pixel (0.0013°) and the red line represents the smoothed average per degree in latitude. (**b**) Map with predicted values of mean Chl for the Finnish lakes. Lake Saimaa is indicated by the symbol ♦ and lake Lappajärvi by ✦.

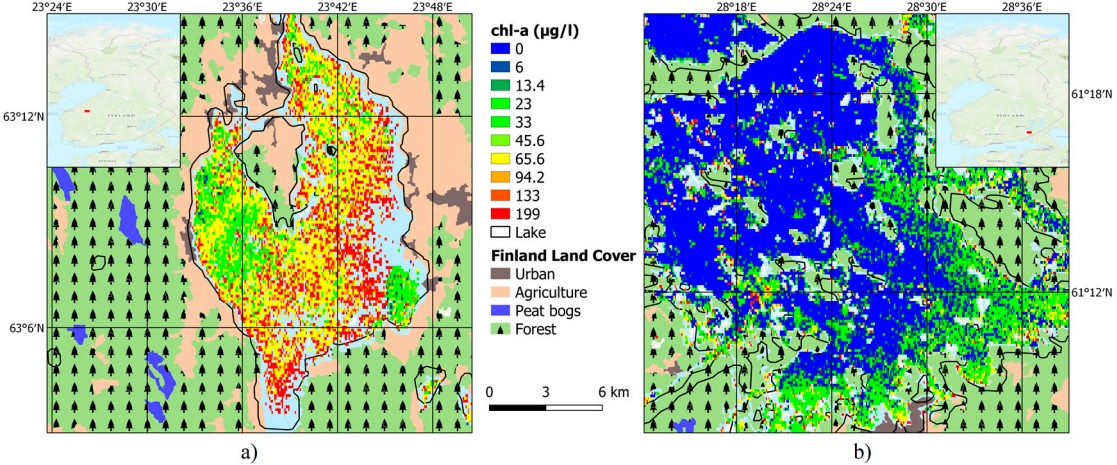

**Figure 14.** (**a**) Lake Lappajärvi is a eutrophic lake surrounded by arable land and some agriculture. (**b**) Lake Saimaa is an oligotrophic lake surrounded by forest and some urban areas to the south. Both maps are provided in the same scale.

## 4. Discussion

In this paper, we aim to better understand how statistical models based on LT perform considering different lakes across an extended time period, from 1984 to mid-2017 and an extended area. There are several differences between the TM and ETM+ sensors, onboard LT 5 and 7, respectively. TM is a multispectral scanning radiometer and ETM+ a whiskbroom scanning radiometer with an additional panchromatic band of 15 m resolution and two 8-bit "gain" ranges. ETM+ also features a 60 m resolution thermal band, replacing the one of 120 m resolution of TM. Nevertheless, these differences do not affect the bands considered in this paper, and we assume we can merge data from both sensors. Regarding the bands used in this study, it should be noted that, as land sensors, most of LT sensing capability (256 gray-levels) is used for the highly reflective surface of the land. Dark lakes are therefore harder to discern, and results are highly dependent on turbidity and trophic levels—benchmarking the differences between these sensing regimes (as the ones in Figure 14) leads to the discussion of

whether using atmospheric correction would lead to different results. Previous results showed that using atmospherically corrected data improved models only slightly [15].

The choice of the spatial buffer around the sampling location has a small effect on the relationship between Chl and satellite data, but a buffer of 500–700 m was found to be the best choice for the models tested in this study. Additionally, a time window of ± 1 day performs better than a ± 2 day, meaning that satellite observations closer to the date when the field samples were collected are preferred. This is in line with previous studies that have assessed Chl over lakes, even in different geographic conditions (e.g., Minnesota, USA) [29]. We also assessed the impact of extending this time-window up to ± 10 days. By doing so, the goodness of predictive Chl estimations has decreased significantly. The same is described by other authors, pointing out the decrease in certainty of the model with a longer time-window [23]. Despite this, our models for non-averaged data performed better than other match-up activities for lakes at lower latitudes. As an example, the best model for a selection of water bodies in Maine, USA provided a correlation coefficient of at most 0.25 [23].

As expected, the use of the SWIR 1 band did not improve the performance of the model derived from intra-annual data. In fact, SWIR bands have been used on the enhancement of the atmospheric correction algorithms for ocean color [37] and their calibration for inland and coastal waters [38]. However, under certain circumstances, the direct use of SWIR bands for Chl detection is fruitful, applicable to floating-bloom events and typical of cyanobacterial communities, where most of the biomass is at surface level [39]. In the oceans, MODIS data from SWIR bands has also been used on the concept of a floating algae index [40]. Our study does not focus on depth-resolved data and thus the use of SWIR was omitted.

On the other hand, our results demonstrate that we can determine, with good confidence, the lake specific mean Chl concentration in Finnish lakes (Figure 12). This general model for all lakes must be used carefully. From Figure 8, we see that the model performance varies amongst the sampled lakes on the daily aggregation approach. However, it is the level of Chl concentration that has the most significant impact on model accuracy. For timely estimations of Chl, the LT bands are not ideal as turbidity has a significant influence on the estimated Chl. Specifically, during the spring months, it was not possible to detect how turbidity is affecting our results. Studies have concluded that turbidity detection was feasible through TM data, but not Chl [17]. It should be noted that if a lake is oligotrophic or CDOM-dominated (like most Finnish lakes) the radiometric resolution of LT data is insufficient to detect different Chl levels—lakes must be eutrophic for signal on both LT5-TM and LT7-ETM+ to be discerned (Figure 9). Authors have argued that high CDOM levels shadow the optical signature of phytoplankton in the blue region of the spectrum rendering the blue/green ratio based Chl algorithms useless [7]. Despite this, we have seen that seasonally aggregating data from satellite and in situ campaigns provides a reliable model. In this model, some concerns must be addressed. For example, the land adjacency effect is particularly relevant in small lakes. As seen in Figure 11c, preliminary results on the performance of the model with respect to how close the sampling locations are to the margins provide insights that need to be further studied.

The adjacency effect is known to reduce apparent surface contrast [41] and is particularly severe in the case of dark water bodies surrounded by dense vegetation. As LT imagery is developed mostly for land applications, there is a mask for the lakes' adjacency effect to land but not the converse. In areas near the lakes' margins, the adjacency effect can overestimate Chl concentration as it acts by changing reflectance at shorter wavelengths [26]. Boreal waters are known for their high concentrations of humic and fluvic acids (CDOM) leached from the surrounding vegetation and soils. These, in turn, reduce the light availability in the blue spectral region [26]. In Finnish lakes, high CDOM absorbs nearly all water leaving radiance. Thus, the radiance measured above lakes consist of radiation backscattered from the atmosphere, radiation reflected from the water surface and the radiation backscattered from the adjacent land. Water signal is often negligible in the blue band and can be discarded for such lakes. It is paramount to have in situ studies that sample a lake at different distances from the coast—such results would greatly improve our capability to quantify the adjacency effect in these types of lakes

where sun elevation is low and most of their waters are brown. From our results, one factor that seems to ameliorate detection performance—including the minimization of the adjacency effect—is the trophic level of the lake. Another evident source of error is the potential effect of shallow lake grounds. Our methodology is robust on defining the lakes' margins by creating a permanent water mask. The use of a buffer away from the lakes margins and small islets can render unlikely the presence of shallow banks under water—nor can we guarantee the avoidance of underwater vegetation. Some of these banks can also appear and disappear over the extensive timespan of this study. The revision of earlier studies on aquatic vegetation could influence resolving phytoplankton from our detection methodology [13].

As seen in lakes Lappajärvi ($\langle$Chl$\rangle$ = 86.95 µg/L) and Saimaa ($\langle$Chl$\rangle$ = 10.1 µg/L), the model correctly identified them as eutrophic and oligotrophic lakes, respectively. Lappajärvi, being a brown water lake, contains humic material and has a high phosphorus content. The phosphorus load into Lappajärvi comes from agriculture and cattle farming, and measures have been considered to decrease the impact of eutrophication and mitigate algae bloom occurrence [42]. Our results also corroborate the trends in eutrophication, such as Lake Keurusselkä ($\langle$Chl$\rangle$ = 31.83 µg/L), reported as having an increasing trophic status, which corresponded to 10 µg/L in 2010 [43]. In Lake Saimaa, depicted in Figure 14b), the average Chl during the summer period is 10.1 µg/L, which also corresponds to the literature, as this is a typical oligotrophic lake [44]. The results of this study are particularly timely now that new data is emerging and Finnish lakes like Kallavesi, Näsijärvi and Vesijärvi have already been reported to experience later freeze dates and earlier ice break-up dates [45]. Understanding the impact of such geophysical changes on lake Chl content is of utmost importance, and can be applied to better understand phenology changes of phytoplankton in freshwater systems [16]. As Figure 15 shows, having a product for Chl can improve environmental management by addressing the surrounding land use changes and their impacts.

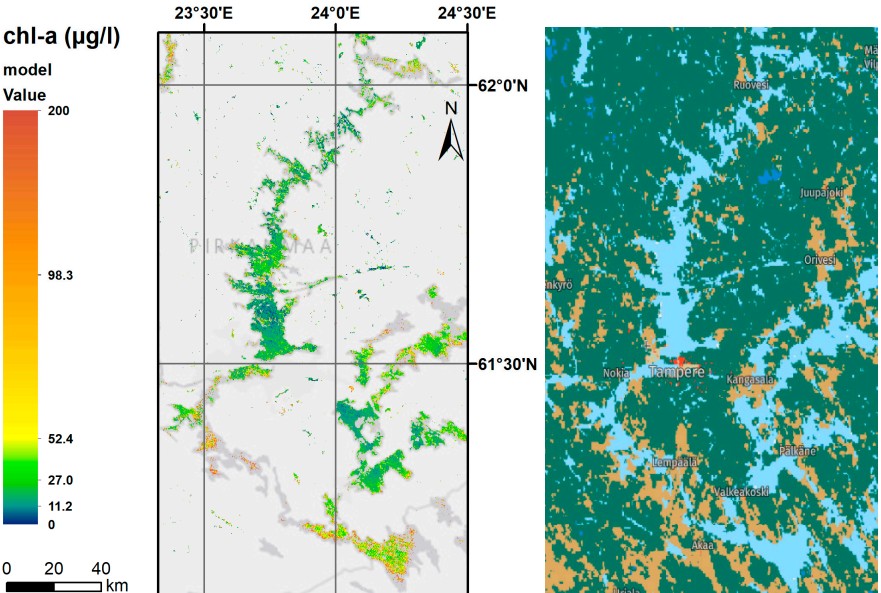

**Figure 15.** Map with predicted values of mean Chl *(left)* for Pirkanmaa lake and corresponding area for land use: light brown indicating arable land or agriculture, dark green represents forest.

Early semi-operative approaches pointed out the possibility of studying small lakes (and their small details, Figure 16) if not for the medium and low resolution of conventional ocean color sensors [14]. Our study also revisits the application of in situ data reference data in the conception of empirical Chl models [12,14,46,47] by applying new cloud computing technology like Google Earth Engine [27] to the analysis of thousands of images and in situ measurements corresponding to long time series of data.

The use of our methodology can allow for the application of a model to smaller lake, more remote, or simply less studied than a bigger "control" lake used for in the validation campaign.

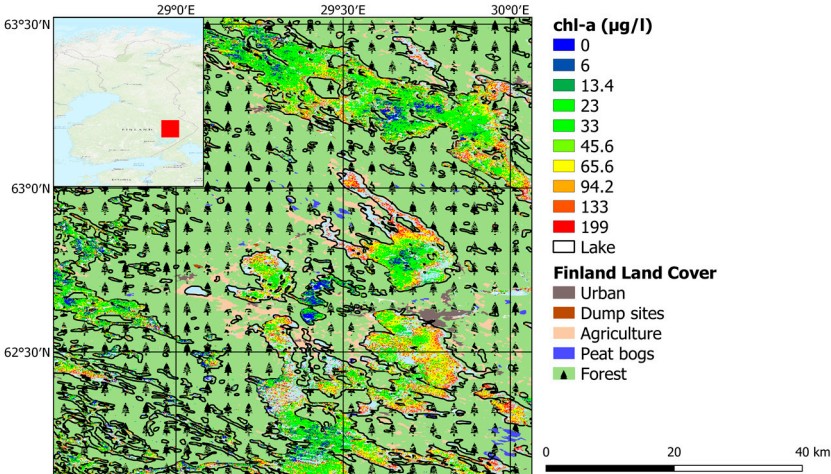

**Figure 16.** Eastern area of Finland with plenty of small lakes surrounded by dense vegetation and thus where the adjacency effect is most prominent. Land cover is simplified: "urban" is discontinuous urban fabric, industrial or commercial units; "agriculture" is taken as agriculture and natural vegetation but also non-irrigated arable land; and "forest" comprises transitional woodland-shrub, broad-leaved forest, coniferous forest, mixed forest and natural grasslands.

Additionally, we underline the acute need to monitor Boreal lakes. In other regions, like Alberta, Canada, variations of trophic levels have been linked to climate, biotic variations and geology [10]. A similar approach based on a model for all Finnish lakes is of the utmost importance.

## 5. Conclusions

Despite its relatively high spatial resolution, LT imagery is often overlooked for the remote sensing of Chl, due to the low spectral resolution of the sensors. Nevertheless, our results demonstrate that estimations over aggregated time windows can be done with high accuracy. Our results show that LT imagery can fill the resolution gap when it comes to monitoring the small, numerous and adjacent lakes found in the Finnish geophysical context.

Calculating representative monthly means for the satellite images can produce useful results of the model allowing for higher correlations and typical characterizations of the different water bodies within specified timeframes. Assessing Chl is possible due to its strong relationship with turbidity, although multi-linear models of LT data to estimate turbidity do not produce results as good as the Chl models. The relationship between Chl and turbidity is most robust during summer, and, since the model performs better during this period, the derived model can be used for further studies of phenology shifts during the blooming seasons.

The adjacency effect, when quantified, could explain the high variations of our model when applied to the calculation of Chl concentration in small lakes. During this study, we came across the difficulty of identifying the impacts of the adjacency effect both through our analysis and the timely literature. Therefore, we believe that the work here can open the discussion for retrieving more sophisticated models for Chl estimation, giving a better suited measurement of adjacency effects from land on small water bodies remotely sensed through LT imagery. Similar studies have already been carried for the adjacency effect on MODIS, SeaWiFS, MERIS, OLCI, OLI and MSI for the case of mid-latitude coastal environments [48]. Although our methodology features a buffer from the lakes' margins, this might not be enough to completely eradicate the impact of the adjacency effect.

Satellites can provide valuable information in cases where required sampling density is high, or field surveys are expensive or even impossible to carry out. Especially, satellite images may provide

informative prior information for field sampling estimates, in a Bayesian setting. As lakes are sensitive to many climate factors, they are certainly useful in climate change monitoring.

**Author Contributions:** F.L., E.E.M. and S.K. conceived the initial idea. E.E.M., F.L. and V.B. designed the methods. F.L., E.E.M. and L.K. collected and processed the data. F.L. and E.E.M. analyzed the results. F.L., V.B., F.D.S., S.K., L.K., E.E.M. contributed to the final version of the manuscript. All authors provided critical feedback and helped shape the research, analysis and manuscript. All authors have read and agreed to the published version of the manuscript.

**Funding:** This research was funded by Portuguese Fundação para a Ciência e a Tecnologia (FCT) grant numbers PD/BD/113932/2015, SFRH/BSAB/142981/201, UIDB/04292/2020 and UIDB/00329/2020. This work was also supported by the European Union's Horizon 2020 Research and Innovation Programme grant agreement N 810139 and the Academy of Finland (decision numbers 318252 and 319905).

**Acknowledgments:** The research grant PD/BD/113932/2015, given by the Portuguese Fundação para a Ciência e a Tecnologia (FCT), funded this work and the authors are thankful for such fundamental support. This project received funding from the European Union's Horizon 2020 research and innovation programme under grant agreement n° 810139. Vanda Brotas received a sabbatical grant from FCT SFRH/BSAB/142981/201. The work carried out in MARE by Vanda Brotas was also funded by FCT grant UIDB/04292/2020. The work carried in cE3c by Filipe Duarte Santos was funded by the FCT grant UIDB/00329/2020. Eduardo Maeda is funded by the Academy of Finland (decision numbers 318252 and 319905). The authors would like to thank the Suomen Ympäristökeskus (SYKE) for the valuable in situ Chl data as well as the constructive criticism provided. We also thank the valuable advice and support from Kari Kallio and Sampsa Koponen. This work was also supported by funding from the European Union's Horizon 2020 Research and Innovation Programme grant agreement N 810139: Project Portugal Twinning for Innovation and Excellence in Marine Science and Earth Observation–PORTWIMS. Finally, the authors would like to thank the Google Earth Engine forum for the help given in producing the tools that are now available for Chl remote sensing validation studies from multiple space platforms.

**Conflicts of Interest:** The authors declare no conflict of interest.

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
