# Peer review of "Spatial Variability and Detection Levels for Chlorophyll-a Estimates in High Latitude Lakes Using Landsat Imagery"

_remotesensing, doi:10.3390/rs12182898_

Round 1
Reviewer 1 Report
The paper is certainly interesting and the work is robust both for the data set and for the methodological procedures used. I believe that with some arrangements the article can be published.
One problem is the error made by the authors in the numbering of the figures which made it very difficult to understand the text relating to the figures.
In the introduction the authors, given the importance, should also add a reference to the MERIS sensor in addition to those made related to SeaWifs and MODIS.
I believe that figure 8 should be created like figures 7 and 9 in a uniform way of information and style. In figure 8 the maps of chl-a and land use do not seem to have the same dimensions.
It is correct to indicate in the adjacency a cause of possible errors but I do not consider it necessary to add the figure of Matthews relating to other aquatic environments, with completely different water quality conditions. Remove the figure.
In evaluating the masking of the images, did the authors consider any disturbance due to the seabed for the shallower areas with high transparency? If yes, indicate in the text, if not, I think it is important to consider it or at least indicate it in the text as a possible source of error.
It is correct to indicate that the high concentrations of CDOM of the boreal lakes can strongly influence the results obtained relating to the Chl-a estimate, but I believe it is necessary to deepen the topic as done for the turbidity. The authors should try to recover data related to CDOM and analyze it.
The authors focus on evaluating the differences in the results due to the spatial temporal comparison, this type of analysis is not new, I think they should increase the number of citations related to other similar studies carried out in lake environments.
Minor comments:
- check the numbering of the figures, there are errors
- check the formatting of the captions of the figures, there are errors (e.g. part of the text in bold and with different styles)
- in row 115 what is chapter 5]?
- in figure 2: change fev with feb, a) and b) is partially covered
- Uniform the acronym of chlorophyll-a, in the tense it is chl a, in the figures chl-a
Reviewer 2 Report
It is a bit strange that a study on remote sensing of Finnish lakes does not mention any lake remote sensing studies carried out in Finland. There is at least 30 years of lake remote sensing history in Finland. Many of the earlier results were published in the lake remote sensing special issue of the Science of the Total Environment published in 2001. Most of the papers were the result of FP4 project SALMON (Satellite remote sensing of lakes) where the Finnish Environment Institute and the Department of Geophysics of University of Helsinki took part. The Department of Geophysics had also nearly 20 years of joint lake studies in Finland and Estonia with scientists from the Estonian Marine Institute. There are several publications based on that cooperation too. There have been several remote sensing Ph.D. theses in Finland (Antti Herlevi, Kari Kallio, Sampsa Koponen, etc.) which are based on the results obtained in Finnish lakes. Many of these published studies use also Landsat data. The Authors are simply not aware what has been done in Finland in the field of lake remote sensing during the last 30-40 years. This demonstrates that the Authors are not very familiar with the field.
The Authors exclude December and January data because low sun elevation. Many Finnish lakes are frozen from October-November to April-May (depending on the year). This kind of indicates that lots of data was collected for the periods when lakes were actually frozen? Later it is said that the data was used only for the period between day 200 and 280 (July 19 to October 7). This means that more than 800 images (according to the Figure 2) were processed and actually not used?). Seems kind of strange. Also other periods (like monthly means) are mentioned later in the study but one cannot find anything about these in the Methods part.
The Authors use coarse water mask while many publicly available global lake water masks exist. There is GLOWABO (Verpoorter et al. 2014) with 14.25 m resolutions. Feng et al (2015) have a global lake water mask with 30 m resolution and there are several others.
The Authors spent time on studying the usefulness of single Landsat bands in retrieving Chl-a. There are tens and tens of this kind of studies from the last 30-40 years. Being familiar with the field would have save lot of time currently spent on useless work.
It is not described in the Methods part what the seasonally aggregated model means. One value for the whole summer? This is an absolutely meaningless number. What is needed is dynamics of Chl-a in each lake during each growing season and then comparing these results between the years. Landsat revisit time is 16 days. With the cloud cover and other issues the realistic revisit time is two months. If there are phytoplankton blooms in lakes then they can emerge, reach their peak and disappear within the 16 days. Meaning that capturing phenology with such low revisit time is almost impossible even if all days are cloud free. In reality, one cannot capture many lake images with Landsat in the latitude of Finland due to cloud cover. So, the difference in summer aggregated data is based on the fact whether one or two images happened to be collected during higher biomass periods or not. So, it is a question of luck not the actual annual differences occurring in lakes. There are monthly means in the Conclusions. Another time scale? Monthly mean should mean one Landsat image in a lucky case and an average of two in extremely lucky case. In how many cases there were actually two images per month to calculate the monthly mean?
The Authors seem not to understand what the adjacency effect means. They say that it lowers water leaving radiance at shorter wavelengths (row 392). This is absolutely not true. The adjacency effect does not change the water leaving radiance by any means. It stays the same whether the adjacency effect exists or not. What is changing is reflectance. At shorter wavelengths, where CDOM absorbs nearly all water leaving radiance in Finnish lakes, the radiance measured above lakes consist of radiation backscattered from atmosphere, radiation reflected from water surface and the radiation backscattered from the adjacent land. The part of water signal is therefore negligible in blue and using the blue band in remote sensing of Finnish lakes (and many other lakes and coastal waters) is useless.
It has also been obvious for many decades that eutrophic lakes are easier to study with remote sensing, especially when suboptimal sensors, like the Landsat, have to be used. First of all, all Landsat sensors (before L8) were 8-bit instruments. Meaning that they have 256 grey-levels to describe the whole variability between absolute black body (like dark brown lakes) and absolutely white surface. Landsat are land sensors. Consequently, they are designed this way that most of these grey-levels are used to describe bright targets like land. The variability between the most black (or blue) water and extremely turbid (blooming) lakes is just 3-4 grey-levels (one of which is instrument noise). Therefore, it is obvious that in dark lakes Landsat cannot detect almost anything and in moderately to extremely turbid lakes it can detect something. This is also the reason why there is no point in atmospheric correction of Landsat data as this just introduces noise (see Kallio et al. 2008 and Kutser 2012).
Most of the turbidity (and suspended matter) in summer is caused by phytoplankton. Thus, it is not surprising that these two are correlated in summer.
Reviewer 3 Report
The detection accuracy of the chlorophyll-a content of high latitude lakes in Finland using Landsat Imagery is investigated. The reviewer would like to thanks the author for carrying out this kind of research which is very meaningful for a better monitoring tool of the climate change. A few comments are listed below for consideration. The reviewer recommended a minor revision without re-review.
- Figure 2: please make sure the images are intact and the abbreviation of month are correct.
- Figure 4: please correct the title of the figure.
- Figure 5: please correct the title of the figure.
- Line 213: Please introduce the full terminology of LMG.
- Line 252-253: Why NDCI shows negative correlation with chlorophyll a (log)? Based on the equation of NDCI, it should be positively correlated with chl-a content. Also, a deeper analysis between log(chl) and log(Bands) is recommended.
- Line 261: Please check the unit of R2 in this sentence.
Round 2
Reviewer 2 Report
Detailed comments
Row 76. Landsat satellites are barely usable for mapping Chl-a. This can be done only in a narrow concentration range not over the whole range of optical variability that occurs in lakes. Thus, speaking about species composition and abundance in the context of Landsat shows poor knowledge in the field.
Row 87. Maximum phytoplankton usually occurs in summer. Not only in oligotrophic lakes but in all lakes. It is strange that the oligotrophic lakes are stressed here. There are probably some lakes that behave similarly to the nearby Baltic Sea where the spring bloom (occurring after ice melt) usually has higher Chl-a than the more spectacular summer blooms. Spring blooms consist of large cells containing lot of Chl-a while summer blooms are caused by cyanobacteria that contain much less Chl-a. Conventional extraction methods also cannot get Chl-a out from cyanobacterial cells. This makes the summer bloom even smaller compared to spring bloom if Chl-a and not cell count is used as a proxy of the bloom intensity.
Row 91. The paragraph starting here shows still very poor knowledge in the field. The first references are about field radiometry and Alpine lakes while there are tens of publications from Finland and other Boreal lakes around the world (Sweden, USA, Canada, etc.). Many of the early studies (from 1980s, 1990s and 2000s) used also Landsat sensors. Thus, there is no point in referencing field radiometry and airborne hyperspectral studies when there are lots and lots of lake Landsat studies available to cite. Some of the stuff is already from pre-pdf era, but a little googling should show plenty of relevant publications.
Row 99. What does it mean “under homogenous environmental conditions”? There are studies on single lakes, but usually they have data from different seasons. In a single lake Chl-a may easily vary by two orders of magnitude over one ice-free season. There are also many studies using tens and hundreds of lakes. You just ignore them.
Row 105. This is the whole point – Landsat is not designed for waterbodies and is very bad sensor to use for such purpose. However, it is the only one having long data series and therefore, it is worth of trying. Landsat Chl-a estimation only works to certain extent over narrow range of lakes. From R2 point of view the worst case is actually to study one relatively stable lake with Landsat. Landsats before L8 have only 3-4 grey levels in each band to describe the whole variability occurring in lake waters i.e. the range of Chl-a between 0.01 and 1000 mg/m3 can be described with four different signal intensities. If the lake is CDOM-dominated (like most Finnish lakes) then increase of Chl-a from 1 to 20 mg/m may not be sufficient to cause Landsat signal to increase by one number. Consequently, the R2 for one single lake may be easily around 0. On the other hand, if there are tens of lakes under investigation and Chl-a varies between 10 and 200 mg/m3 then the R2 may be quite good even if the relative error in Chl-a estimates is 200-500%. Thus, the R2 does not depend much on the algorithm/method you use, but depends more on what kind of lakes you have under investigation.
Row 106. This statement is simply wrong as was already stressed earlier.
Row 126. Many Finnish lakes, and Boreal lakes in general, contain so high amount of CDOM that the CDOM absorption affects strongly all visible wavelengths, not only the blue band. In black lakes CDOM absorption at 700 nm may still be higher than absorption by water molecules (that is huge at these wavelengths). This is the reason why Landsat’s (and all other remote sensing sensors) blue band(s) are practically useless in Boreal lakes (and many coastal waters). The only thing the blue bands detect is atmosphere and glint as there is no water leaving signal.
Row 137. The Introduction stresses many times the need to study small lakes and yet, you actually study the largest lake in Finland and several other relatively large lakes. Thus, the introduction has to stress more geomorphologically complex lakes rather than small lakes as only a few small lakes were studied in this paper. It is also not clear why these lakes were selected. Finnish lake monitoring database contains data from many more lakes in the area.
Row 215. Landsat does not have myriad of band combinations. It has a very few (3) water penetrating bands that can give us information about water Chl-a content. Some NIR bands may be useful if there are cyanobacterial blooms in the water or cyanobacterial scum floating on the water surface. On the other hand, the blue band is useless in most cases as most of Finnish lakes are CDOM-dominated and there is very little or no water leaving signal in blue.
Row 244. You use BRG developed for a very blue oligotrophic lake (Garda) in CDOM-dominated red to black lakes. This does not make much sense. On the other hand, the BRG is basically green peak height algorithm against blue-red baseline. Thus, the logic of choosing it is wrong but the algorithm itself may be meaningful also in Finland. There are several Landsat studies for Finland and other Boreal lakes That you don’t cite). Why the most promising algorithms were not taken from there?
Row 268. There is no water leaving signal in band 5. Using it was just waste of time from the beginning. Read water remote sensing literature.
Row 312. Again, this has been known for decades that there is enough signal to detect Chl-a with Landsat only if lakes are eutrophic and there is some signal to measure. If lakes are CDOM-dominated (like most of lakes in Finland) or oligotrophic and very clear blue, then Landsats (before L8) simply do not have sensitivity to detect anything useful.
Row 389. Using of SWIR bands in aquatic remote sensing is a bit stupid. There is a way to use NIR or SWIR bands, but you did not do this. Namely, subtraction NIR or SWIR band value from all visible bands (that do actually contain information about water properties) removes the residual errors of atmospheric and/or radiometric corrections as well as glint and haze. For example, a proper way of using BRG would have been ((b1-b4)-(b3-b4))/(b2-b4). Again, a proper way to do research is to learn at least a bit about the field and then do something meaningful not to repeat things that have been known for decades. The lack or relevant references in the Introduction shows that you are not familiar with the field and using the SWIR bands stresses it even further. As a reviewer I see this more and more that researchers use GEE and different AI tools without actually knowing much (or sometimes anything) about the field. A lots of useless work is done and massive amount of meaningless results are produced this way.
Row 447. You stress small lakes again whereas most of the lakes you actually study are quite big. If you can afford having 200-700 m buffer zones then the lakes have to be very large. Small lakes have 200-700 m (or less) diameter.
